# Phase Angle as a Predictor of Mortality in Older Patients with Hip Fracture

**DOI:** 10.3390/nu16142221

**Published:** 2024-07-11

**Authors:** Francisco José Sánchez-Torralvo, Verónica Pérez-del-Río, Luis Ignacio Navas Vela, María García-Olivares, Nuria Porras, Jose Abuín Fernández, Manuel Francisco Bravo Bardají, David García de Quevedo, Gabriel Olveira

**Affiliations:** 1Unidad de Gestión Clínica de Endocrinología y Nutrición, Hospital Regional Universitario de Málaga, 29007 Malaga, Spain; fransancheztorralvo@gmail.com (F.J.S.-T.); gabrielm.olveira.sspa@juntadeandalucia.es (G.O.); 2Instituto de Investigación Biomédica de Málaga (IBIMA), Plataforma Bionand, 29010 Malaga, Spain; 3Departamento de Medicina y Dermatología, Facultad de Medicina, University of Malaga, 29010 Malaga, Spain; veronicaperezdelrio@gmail.com; 4Unidad de Gestión Clínica de Cirugía Ortopédica y Traumatología, Hospital Regional Universitario de Málaga, 29010 Malaga, Spain; 5Centro de Investigación Biomédica en Red de Diabetes y Enfermedades Metabólicas Asociadas (CIBERDEM), Instituto de Salud Carlos III, 28029 Madrid, Spain

**Keywords:** bioelectrical impedance analysis, hip fractures, mortality, aged, phase angle, muscle mass

## Abstract

The aim of our study is to determine if there is an association between phase angle obtained by bioelectrical impedance analysis (BIA) and mortality in older patients with fragility hip fractures. A prospective study of patients over 65 years old and hospitalized with a diagnosis of hip fracture was conducted. BIA was performed 24 to 48 h after surgery. Mortality was recorded, and the optimal phase angle cut-off value for predicting mortality was determined by using receiver operating characteristic (ROC) curves. A total of 262 patients were included. Of the patients studied, 10 (3.8%), 21 (8%), 39 (14.9%) and 53 (20.2%) died at 1, 3, 6 and 12 months after surgery, respectively. The phase angle cut-off for mortality at 12 months was 4.05° in women and 4.65° in men. A total of 94 patients (35.9%) were considered to have a low phase angle. After adjustment for possible confounders, mortality in patients with a low phase angle was 5.1 times higher at 1 month, 3.1 times higher at 3 months, 2.9 times higher at 6 months, and 2.8 times higher at 12 months. Phase angle is associated with prognosis in patients admitted for hip fracture regardless of age and comorbidities and can be positioned as a prognostic tool for mortality at 1, 3, 6 and 12 months.

## 1. Introduction

With the increasing incidence of osteoporosis in the older population, there is a need to develop strategies for the prevention and management of fragility fractures. These are caused by low-impact trauma and affect mainly the humerus, wrist, vertebrae and hip [1], the latter being associated with a significant risk of mortality and refracture, with the subsequent economic costs [2,3]. Worldwide, there are approximately 1.7 million cases of hip fracture per year [4], of which 620,000 occur in Europe [5].

Several situations and factors have been found to be associated with mortality in patients with hip fracture, including malignancy, diabetes, pulmonary and cardiovascular disease [6], COVID-19 [7], dementia [8], time to surgery [6,9], underweight and malnutrition [10,11].

However, mortality after hip fracture has decreased over the last 60 years, probably due to a multidisciplinary management approach that focuses on both prevention and treatment, as well as the different circumstances surrounding the patient [12]. In this context, Fracture Coordination Units (FCU or FLS, Fracture Liaison Services), whose activities focus on the secondary prevention of fragility fractures following a multidisciplinary approach [13], have demonstrated their usefulness since 2011 [14], with a significant reduction in all-cause mortality [15].

Nutrition plays an important role in this prevention, since a positive association has been found between malnutrition and the incidence of hip fractures [16]. Malnutrition is associated with a higher incidence of complications, poorer functional recovery and higher mortality. Nutritional status is also associated with morbidity and mortality [16,17,18]. A variety of definitions of malnutrition have been proposed, and body composition has become increasingly important over the years. For example, the Global Leadership Initiative on Malnutrition (GLIM) included reduction in muscle mass, assessed by a validated method of measurement, as one of the phenotypic criteria proposed to diagnose malnutrition in 2018.

These methods include bioelectrical impedance analysis (BIA) [19]. BIA directly measures bioelectrical parameters and uses prediction equations to estimate the mass of different body compartments [20,21]. However, this estimation can be influenced by multiple factors and depends on body mass [20,21], which is often difficult to measure in recently operated patients. BIA also provides the phase angle (PhA), an index derived from raw bioelectrical parameters that is considered to reflect cell integrity and function [20,21]. PhA is positively associated with tissue reactance (Xc), which is associated with cell mass, integrity, function, and composition, and is negatively associated with resistance (R), which depends mainly on the degree of tissue hydration. PhA is calculated by the following formula: PhA = arc tangent (Xc/R) × 180°/π) [20]. PhA is a raw parameter of cellular health, equivalent to the “electrocellgram^®^”. It has prognostic value in health and disease. As an isolated numerical value, it provides global and non-specific information about cell mass and health, as well as congestion or inflammation [21]. A standardized PhA (SPhA) can be calculated using the reference values matched for sex and age by subtracting the reference PhA from the patient’s observed PhA and then dividing the difference by the standard deviation of the respective age and sex references [22].

The PhA is considered a good predictor of mortality in a variety of clinical conditions [22,23,24,25]. Whereas several studies have examined and demonstrated the association between muscle mass and mortality in older people with hip fracture [26,27,28], we found no studies analyzing the relationship between phase angle and mortality in these patients. Nonetheless, it has been associated with a higher risk of fracture [29] and poorer function after rehabilitation [30].

Our hypothesis is that a low PhA is associated with mortality in patients undergoing surgery for a fragility hip fracture.

The aim of our study is to determine the association between BIA-determined PhA and mortality at 1, 3, 6 and 12 months after a fragility hip fracture in older patients.

## 2. Materials and Methods

Prospective study of patients over 65 years of age hospitalized and with a diagnosis of hip fracture at the Trauma Surgery Unit of the Regional Hospital of Malaga, between September 2019 and February 2021.

A flow diagram of the design of the study is reflected in Figure 1.

The type of fracture and the presence of a previous fracture were recorded. Medical comorbidities were measured using the Charlson Comorbidity Index (CCI), which establishes a score that increases with the 10-year risk of death given by a patient’s comorbid diseases [31]. Pre-fracture functional status was assessed using the Barthel Index, which measures a person’s ability to care for himself (ranging from 0 to 100, with 100 the score for an independent person) [32], and the Functional Ambulation Category Scale (FAC), which measures the amount of manual assistance that a person needs for ambulation (ranging from 0 to 5, with 5 the score for an independent ambulator) [33]. C-reactive protein (CRP), prealbumin and albumin were measured in serum by using the following assay kits: Atellica^®^ CH Albumin (Alb) and Atellica^®^ CH Albumin BCP (AlbP) for albumin, Atellica^®^ CH Prealbumin (PreAlb) for prealbumin, and Atellica^®^ CH C-Reactive Protein_2 (CRP_2) for CRP. The CRP/albumin ratio was calculated.

Body composition was assessed in the first 24–48 h after the surgery.

When possible, height was determined by using a stadiometer (Holtain Limited, Crymych, UK), and body mass was determined by using a scale set to 0.1 kg (SECA 665, Hamburg, Germany). When height and body mass could not be determined, patient-reported data were used.

BIA was performed with the Akern BIA-101/Nutrilab analyzer (Akern SRL, Pontassieve, 160 Florence, Italy). Measurements were taken in a comfortable area in the morning, with the patients fasting, and were performed by the same operator in all cases. Post-surgical hydration protocols had been followed correctly in all cases. Possible interferences of solutions and other intravenous products were avoided at the time of the assessment. Patients adopted a supine position for at least 5 min, with the upper (30°) and lower (45°) limbs abducted [29,30]. Electrodes were placed on the hand and ipsilateral foot, which had been previously cleaned. Software (AKERN Bodygram Dashboard, Pontassieve, Florence, Italy) was used to determine the following parameters, which were calculated automatically and displayed on the device screen: PhA, SPhA, body cell mass (BCM), fat mass (FM), fat-free mass (FFM), fat-free mass index (FFMI) and appendicular skeletal muscle mass (ASMM).

For FFMI, the cut-off points established by ESPEN were applied, with low FFMI being considered for values <15 kg/m^2^ in women and <17 kg/m^2^ in men [34].

For ASMM, the cut-off points established by Studenski [35] and recommended by the European Working Group on Sarcopenia in Older People [36] were used, considering low values <20 kg in men and <15 kg in women.

After discharge, patients were followed telematically (by reviewing their medical history), and mortality was recorded at 1, 3, 6 and 12 months.

Quantitative variables were expressed as mean ± standard deviation. Comparison between qualitative variables was performed using the chi-squared test, with Fisher’s correction when necessary. Comparison between quantitative variables was performed using the Student’s *t*-test for variables following a normal distribution and non-parametric tests (Mann–Whitney or Kruskal–Wallis) for those which did not.

Evaluation of the performance of PhA in predicting mortality was based on receiver operating characteristic (ROC) curves and area under the curve (AUC). We estimated the accuracy of these measurements using AUC by plotting sensitivity versus specificity. ROC curves were used to determine optimal cut-off values by finding the point of convergence for the greatest sensitivity and specificity.

Multivariate logistic regression models were used to assess the association between mortality and low PhA, controlling for sex, age and Charlson comorbidity index, since these were the variables which were associated with mortality at 1, 3, 6, and 12 months according to the multivariate analysis. For calculations, significance was set at *p* < 0.05 for two tails.

Data analysis was performed using SPSS 22.0 (SPSS Inc., Chicago, IL, USA, 2013).

## 3. Results

A total of 262 patients were included (Figure 1). The general characteristics of the sample are shown in Table 1.

BIA showed a PhA of 5.18 ± 1.13° in men and 4.5 ± 0.94° in women, a statistically significant difference (*p* < 0.001). The FFMI was 20.9 ± 9.6 kg/m^2^ in men (9.1% below 17 kg/m^2^) and 17.6 ± 2.1 kg/m^2^ in women (8.7% below 15 kg/m^2^). The ASSM was 20.9 ± 3.6 kg for men (38.2% below 17 kg) and 15.1 ± 2.6 kg for women (52.7% below 15 kg). The other BIA composition parameters are shown in Table 2.

During follow-up, the cumulative number of patients who died from the time of surgery was 10 patients (3.8%) in the first month, 21 patients (8%) in the first three months, 39 patients (14.9%) in the first six months, and 53 patients (20.2%) in the first twelve months.

Using the ROC curve, we determined the PhA cut-off points for predicting mortality at 12 months (Figure 2). The ROC curve analysis showed that PhA had a discriminatory ability to detect mortality. The PhA cut-off for mortality at 12 months was 4.05°, AUC = 0.674 (sensitivity 69.7% and specificity 59.7%) in women and 4.65°, AUC = 0.693 (sensitivity 72.7% and specificity 63.6%) in men.

Considering the above cut-off points, 94 patients (35.9%) were considered to have a low PhA.

A comparison of general characteristics, body composition and biochemical parameters according to the PhA of the patients is shown in Table 3.

Multivariate analysis showed an increased risk of mortality at 1, 3, 6 and 12 months among patients with a low PhA. An association was also found between age, sex and the Charlson Comorbidity Index, and mortality at 1, 3, 6, and 12 months (*p* < 0.001 at all times). Therefore, these variables were included in the logistic regression adjustment. Table 4 shows the relationship between mortality at 1, 3, 6 and 12 months and low PhA, adjusted for age, sex and Charlson Comorbidity Index.

## 4. Discussion

In our study, PhA was associated with prognosis in patients admitted for hip fracture, independent of age and comorbidities. To our knowledge, this is the first study to investigate and establish an association between the PhA and mortality in patients with hip fracture.

Malnutrition is associated with increased mortality in older patients with hip fracture [11]. Nutritional intervention is cost-effective and has been associated with improved nutritional status and a greater functional recovery [18]. The use of BIA as a tool to assess nutritional status has increased in recent years and has been included in some protocols for the assessment of older patients undergoing surgery for hip fracture [37,38,39]. These protocols include BIA parameters that reflect muscle mass, such as the skeletal muscle index (SMI) or ASMM.

Muscle mass and mortality are closely related in these patients. Low muscle mass, as defined by a low SMI, predicts higher one-year mortality [10,27]. Furthermore, ASMM measured by dual-energy X-ray absorptiometry correlates with mortality [26]. In our sample, association was found between mortality and malnutrition using the BIA as a determinant of muscle mass according to the GLIM criteria [11]. Our hypothesis is that the use of BIA-estimated FFM may not be as good a predictor as PhA itself, because focusing on raw parameters such as PhA eliminates the bias of using prediction equations based on benchmark models.

In our study, patients with a low PhA had lower FFM and ASSM. Low PhA is associated with progression of frailty and sarcopenia, disability and poor outcomes in geriatric patients [21]. PhA is inversely related to muscle mass and strength in older people and may be a good bioelectrical marker to identify older patients at risk of sarcopenia [40]. Sarcopenic hip fracture patients had a higher mortality rate than those with normal musculature [26]. In this regard, the patients in our sample with low PhA had higher comorbidity (represented by a lower Charlson Comorbidity Index) and worse functional status (represented by a lower Barthel Index).

In a study of older women, improvements in the muscle quality index induced by resistance training were associated with increases in PhA [41]. Therefore, identifying older patients with low PhA and implementing an exercise program could lead to an improved prognosis in those who suffer a hip fracture.

There is increasing evidence that dynamic factors such as inflammation and oxidative damage may directly influence PhA levels in conditions that affect cell membrane integrity and function [42]. In our study, patients with low PhA had higher inflammatory parameters and lower BCM. The inflammatory process underlying the hip fracture and its intervention, as well as common situations in these patients, such as malnutrition, sarcopenia or cachexia, lead to a net loss of BCM and cell membrane surface area, which in turn leads to a decrease in the PhA [21].

A previous study found that higher mortality in patients with hip fracture was associated with lower reactance [43], a component of the PhA related to cell mass, integrity, and function [44]. PhA is a parameter that is sensitive to changes over time and may therefore be useful in evaluating different treatments [21]; it has been associated with mortality in many other clinical conditions, and cut-off points have been identified to predict mortality in these settings [23]. We are the first to report these cut-off points in older patients with hip fracture.

The strengths of this prospective study lie in the number of subjects included and the medium-term follow-up. In addition, the simple and standardized technique for estimating prognosis may justify its use in a variety of clinical settings, both inpatient and outpatient, for a range of pathologies.

There are also some potential limitations. As a single-center observational study, the results must be interpreted with caution, without establishing causal relationships. The fact that patients underwent hip surgery hours before the assessment may have affected the PhA measurements, making it more difficult to distinguish the inflammatory component from cell mass loss or cell damage.

## 5. Conclusions

PhA is associated with prognosis in patients admitted for hip fracture, independent of age and comorbidities, and can be positioned as a prognostic tool for mortality at 1, 3, 6, and 12 months. Measures based on this knowledge could lead to an improved prognosis in these patients.

## Figures and Tables

**Figure 1 nutrients-16-02221-f001:**
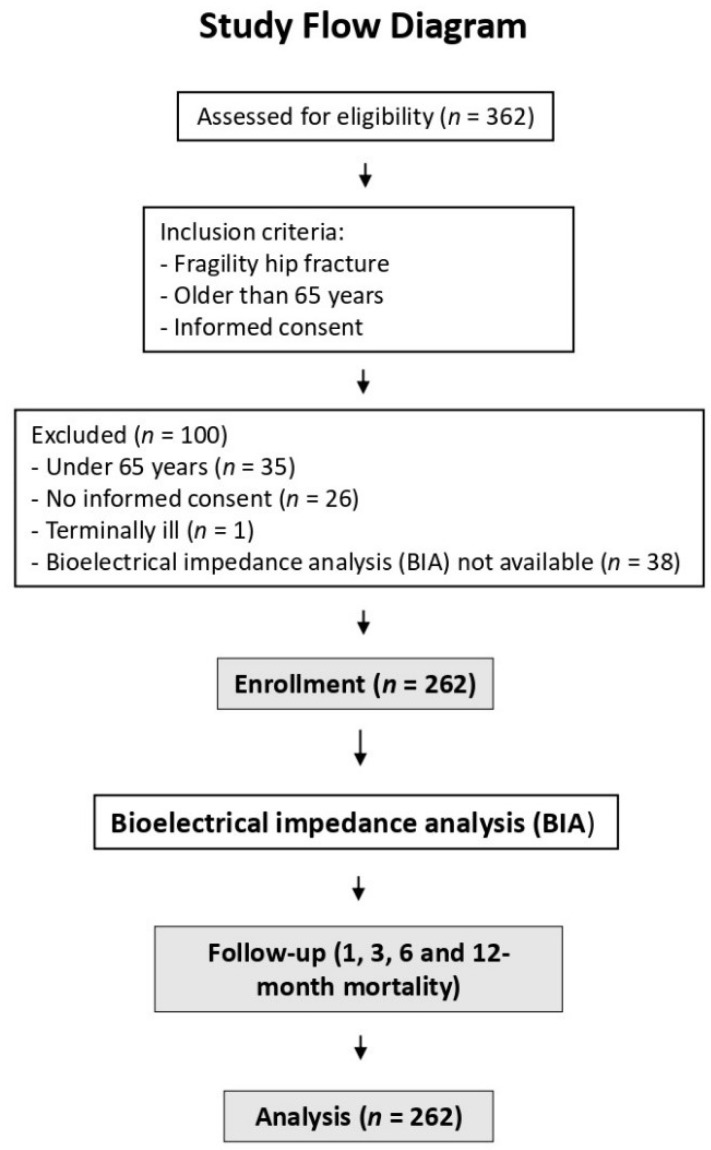
Study flow diagram and research methodology.

**Figure 2 nutrients-16-02221-f002:**
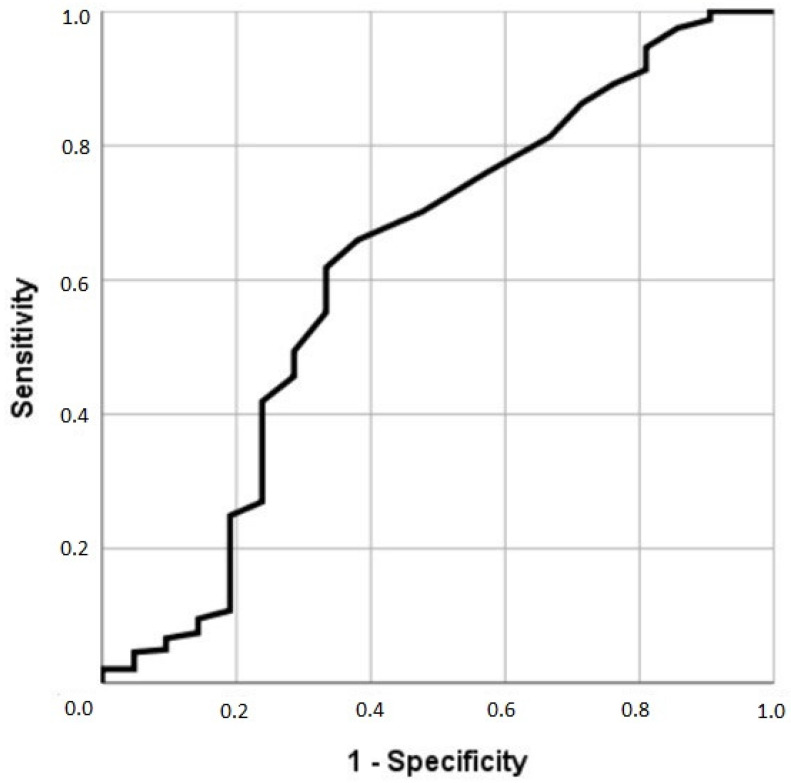
ROC-curve analyses for phase angle to predict mortality.

**Table 1 nutrients-16-02221-t001:** General features.

	*n* = 262
Age (years) †	82.9 ± 7.1
Sex *	
Men	55 (21)
Women	207 (79)
Charlson Comorbidity Index †	5.63 ± 1.9
Barthel Index †	74.69 ± 27.33
Functional Ambulation Category Scale *	
0	67 (25.8)
1, 2, 3	182 (70)
4, 5	11 (4.2)
Type of fracture *	
Pertrochanteric	117 (44.7)
Subcapital	114 (43.5)
Subtronchanteric	15 (5.7)
Basicervical	15 (5.7)
Transcervical	1 (0.4)
Previous fracture *	30 (11.5)
C-reactive protein (mg/L) †	115.1 ± 56.7
Albumin (g/dL) †	2.5 ± 0.4
Prealbumin (g/dL) †	13.4 ± 5
C-reactive protein/Albumin ratio†	46.6 ± 25.3
Length of stay †	8.1 ± 5.6
1-month exitus *	10 (3.8)
3-month exitus *	21 (8)
6-month exitus *	39 (14.9)
12-month exitus *	53 (20.2)

* These variables are displayed as n, which refers to the total of individuals that meet the characteristic, followed by the percentage between brackets. † These variables are displayed as mean ± standard deviation.

**Table 2 nutrients-16-02221-t002:** Body composition parameters (BIA).

Parameters	Men (*n* = 55)	Women (*n* = 207)	*p* Value
Body mass index (kg/m^2^)	26.3 ± 3.4	25.6 ± 5.3	0.038
Phase angle (°)	5.18 ± 1.13	4.5 ± 0.94	<0.001
Standardized phase angle (°)	−0.68 ± 0.71	−0.66 ± 0.85	0.895
Body cell mass (kg)	28.6 ± 6.6	19.6 ± 4.2	<0.001
Fat mass (kg)	18.9 ± 8.1	20.2 ± 9.9	0.326
Fat-free mass (kg)	57.6 ± 7.8	42.9 ± 5.4	<0.001
Fat-free mass index (kg/m^2^)	20.9 ± 9.6	17.6 ± 2.1	<0.001
Appendicular skeletal muscle mass (kg)	20.9 ± 3.6	15.1 ± 2.6	<0.001

Data are presented as mean ± standard deviation. BIA: bioelectrical impedance analysis.

**Table 3 nutrients-16-02221-t003:** Characteristics and nutritional parameters according to PhA.

	Normal PhA (*n* = 168)	Low PhA (*n* = 94)	*p* Value
Age (years)	81.7 ± 7.2	84.9 ± 6.4	<0.001
Charlson Comorbidity Index	5.3 ± 1.7	6.3 ± 2	<0.001
Barthel Index	80.8 ± 24.5	63.8 ± 28.9	<0.001
Length of stay	7.8 ± 4.4	8.5 ± 7.3	0.324
Body mass index (kg/m^2^)	26.1 ± 4.6	25.1 ± 4.6	0.128
Body cell mass (kg)	23.5 ± 6.1	17.9 ± 3.9	<0.001
Fat-free mass (kg)	47.1 ± 8.9	43.8 ± 7.2	0.003
Fat-free mass index (kg/m^2^)	18.8 ± 5.6	17.2 ± 2	0.010
Appendicular skeletal muscle mass (kg)	16.9 ± 3.8	15.4 ± 3.3	0.001
Albumin (g/dL)	2.6 ± 0.4	2.5 ± 0.5	0.037
Prealbumin (g/dL)	13.9 ± 4.3	12.5 ± 6.1	0.046
C-reactive protein (mg/L)	106.4 ± 52.8	130.9 ± 60.3	0.002
C-reactive protein/albumin ratio	42.3 ± 23.6	54.3 ± 26.7	0.001

The variables are shown as mean ± standard deviation. PhA: phase angle.

**Table 4 nutrients-16-02221-t004:** Relationship between mortality and low PhA, adjusted for age, sex and Charlson Comorbidity Index.

	*n* = 262	Crude				Adjusted			
		Odds Ratio	95% CI		*p* Value	Odds Ratio	95% CI		*p* Value
			Lower	Upper			Lower	Upper	
Low Phase Angle									
1-month mortality		7.72	1.6	37.16	0.011	5.11	1.02	25.78	0.048
3-month mortality		4.03	1.56	10.37	0.004	3.12	1.16	8.36	0.024
6-month mortality		3.49	1.73	7.07	<0.001	2.85	1.37	5.95	0.005
12-month mortality		3.61	1.93	6.75	<0.001	2.83	1.47	5.43	0.002

CI: confidence interval; PhA: phase angle.

## Data Availability

The datasets presented in this article are not readily available due to technical limitations.

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
