# Peer review of "Phase Angle as a Predictor of Mortality in Older Patients with Hip Fracture"

_nutrients, 2024, doi:10.3390/nu16142221_

Round 1

Reviewer 1 Report

Comments and Suggestions for Authors

This is interesting to find that phase angle is associated with prognosis in patients admitted for hip fracture. Although intriguing, I also question its practicality. I have a few questions for clarification.

Major:

1. Method: Lines 126-127: “controlling for sex, age, and Charlson comorbidity index” – Why did you consider only these factors when discussing the impact of phase angle on mortality? For example, the authors mentioned in Lines 176-178 that muscle mass significantly affects mortality. Since this study measured muscle mass (e.g., ASMM), why was it not included as a control variable?

2. Results: The results section presents numerous biochemical data, such as CRP, and Table 3 shows significant differences between normal PhA and low PhA. However, these data were not controlled for when predicting mortality. The purpose and relevance of presenting these data in relation to this study need more detailed explanation.

3. Results: Lines 145-146: The ROC curve analysis showed that PhA had a “significant” discriminatory ability to detect mortality. How was this significance demonstrated? An AUC of 0.674 or 0.693 is not considered a very strong performance.

Minor:

1. In the Materials & Methods section, it is not mentioned how the phase angle values were obtained. Did the BIA machine measure and display this value directly, or was it obtained in another way?

2. Lines 123-125: “ROC curves were used to determine optimal cut-off values by finding the point of convergence for the greatest sensitivity and specificity.” Did you use the Youden Index for this?

3. Results: Lines 134-135: “BIA showed a PhA of 5.18 ± 1.13º in men and 4.5 ± 0.94º in women, a statistically significant difference (p < 0.001).” Since you performed a test for the phase angle difference between men and women, why not do the same for other variables? It would be beneficial to add another column and include tests for other variables as well.

4. Discussion: Line 174: “musculoskeletal index (SMI)” – do you mean skeletal muscle index or skeletal muscle mass index?

5. Discussion: Lines 199 & 201: The full name of BCM is not provided, only the abbreviation.

Comments on the Quality of English Language

NO

Author Response

Thank you for your feedback on our article "Phase angle as a predictor of mortality in older patients with hip fracture". I will be responding to your comments in this letter, giving reasons for our decisions and explaining the changes made in response to your suggestions. When referring to a line in the article, I will use the numbers corresponding to the new and revised version of the manuscript.

Major:

Comments 1: Method: Lines 126-127: “controlling for sex, age, and Charlson comorbidity index” – Why did you consider only these factors when discussing the impact of phase angle on mortality? For example, the authors mentioned in Lines 176-178 that muscle mass significantly affects mortality. Since this study measured muscle mass (e.g., ASMM), why was it not included as a control variable?

Response 1: Thank you for pointing this out. As mentioned in lines 181-183, age, sex and Charlson Comorbidity Index were included in the logistic regression adjustment because they were associated with mortality at 1, 3, 6, and 12 months in the multivariate analysis. Sex was inadvertently left out of the text of the first version of the manuscript, we have included it in the new one. On the other hand, neither fat-free mass nor appendicular skeletal mass were independently associated with an increased risk of mortality. To make this clearer, a statement has been added to the Materials and methods section (lines 144-146).

Comments 2: Results: The results section presents numerous biochemical data, such as CRP, and Table 3 shows significant differences between normal PhA and low PhA. However, these data were not controlled for when predicting mortality. The purpose and relevance of presenting these data in relation to this study need more detailed explanation.

Response 2: The goal of presenting these analytical parameters was to show the difference in the inflammatory state between patients with a normal and low phase angle according to our cutoff points. As stated in our response to your previous comment, there were no relation between variables in table 3, save from age and Charlson Comorbidity Index, and mortality in the multivariate analysis. Therefore, they were not included in the logistic regression adjustment.

Comments 3: Results: Lines 145-146: The ROC curve analysis showed that PhA had a “significant” discriminatory ability to detect mortality. How was this significance demonstrated? An AUC of 0.674 or 0.693 is not considered a very strong performance.

Response 3: The use of the term "significant" in those lines was intended to emphasise the statistically significant association between low phase angle and mortality at 1, 3, 6 and 12 months after hip fracture surgery, as shown in Table 4. As we agree that its use may be somewhat misleading, the term has been suppressed in the revised version of the manuscript. 

Minor:

Comments 1: In the Materials & Methods section, it is not mentioned how the phase angle values were obtained. Did the BIA machine measure and display this value directly, or was it obtained in another way?

Response 1: Phase angle was displayed directly by the BIA device. In response to this comment, a list of parameters calculated and displayed by the BIA machine has been added to the “material and methods” section (lines 121-125). Additionally, a brief explanation of the method of calculation for phase angle and standardized phase angle has been added to the introduction (lines 68-71 and 74-78).

Comments 2: Lines 123-125: “ROC curves were used to determine optimal cut-off values by finding the point of convergence for the greatest sensitivity and specificity.” Did you use the Youden Index for this?

Response 2: The Youden Index was obtained manually, since the version of the software used for the statistics did not provide it.

Comments 3: Results: Lines 134-135: “BIA showed a PhA of 5.18 ± 1.13º in men and 4.5 ± 0.94º in women, a statistically significant difference (p < 0.001).” Since you performed a test for the phase angle difference between men and women, why not do the same for other variables? It would be beneficial to add another column and include tests for other variables as well.

Response 3: We agree with this comment. Therefore, we have redesigned Table 2 to show the statistical differences between sexes that the parameters have.

Comments 4: Discussion: Line 174: “musculoskeletal index (SMI)” – do you mean skeletal muscle index or skeletal muscle mass index?

Response 4: We meant “skeletal muscle index”. We have changed the term in the revised version of the manuscript.

Comments 5: Discussion: Lines 199 & 201: The full name of BCM is not provided, only the abbreviation.

Response 5: In response to a previous comment of yours, we have added the term to the material and methods section of the revised version of the manuscript, together with the rest of parameters displayed by the BIA device and which had not been mentioned previously (lines 123-125).

We hope that the answers given and the changes made are appropriate and accurate.

Reviewer 2 Report

Comments and Suggestions for Authors

Overview

The authors conducted a prospective observational study.  Phase angle using BIA was measured as a representation of nutritional status (i.e., fat-free mass), and evaluated as a predictor of morality in patients having surgery for fragile hip.  Mortality was quantified at 1, 3, 6, and 12 mo post surgery.  BIA was performed Twenty-four to 48 hours post-surgery, subjects were measured for phase angle, which was used to determine components of body composition.  Receiver operator curve analysis was used to categorize subjects into normal phase angle and low phase angle.  Among 262 participants, 10 died at one mo post-surgery, 21 passed at three mo, 39 at six mo, and 53 at 12 mo.  Blood proteins related to nutritional status or inflammation, standard indexes, and body composition components were statistically different between those categorized as having normal phase angle and low phase angle. With accounting for sex, age, and co-morbidities, odd ratios for mortality with low phase angle spanned from 2.8 to 5.1 for the longest and shortest post-surgery interval.  Phase angle was concluded to be a potential factor for predicting the risk of mortality in this population (those having surgery for hip fracture).

Major Concerns

The authors claim that no prior study has investigated the relationship between mortality and phase angle in those with hip fractures (line 72 and lines 166-167).  While that might be true for phase angle, 145 references came up in PubMed using a search on the words, mortality, hip fracture, elderly, and muscle.  For some of these studies, the measurement of muscle was more direct or specific than phase angle.  The authors have acknowledged the benefit of phase angle (i.e., unobtrusive and doesn’t require body mass assessment) but I think they could improve their acknowledgment of what has been done previously.

Because of the longitudinal aspect, i.e., following mortality periodically over 12 mo, it isn’t clear whether the authors adjusted mortality rates for prior deaths.  If not, a survival analysis might be warranted.

Specifics Comments/Questions 

Lines 23-24: Here or in a more appropriate place in the abstract, add that receiver operating curves were used to determine cutoffs for mortality.

Line 33, Key Words: Add “muscle mass” as a key phrase.

Lines 93-94: Were the concentrations of these proteins measured in serum or plasma?  Please specify the methods.  Perhaps a reference or two or the source of the assay kits could be used here.

Lines 101-107:  Please indicate/add how the researchers ensured patients arrived well hydrated and how long they remained in the supine position.  These factors impact accuracy or reliability of the BIA measurements.  With the assessment done 24 to 48 h post-surgery, a time when subjects may be in negative nitrogen balance, when any requirements for BIA use violated?

Lines 105-108: How was phase angle converted to fat-free mass?  In other words, what equation or constant was used to go from phase angle to fat-free mass?  Was the equation valid for this age group of males and females?  Again, a reference to the validation might suffice here.  What is the sample size for FFM and muscle mass?  Line 97 states “When possible, height was determined by using a stadiometer.  How many were unable to participate in the stature measurement?

Line 133, Table 1: The numbers seem to be misaligned for the column headings or perhaps I’m misinterpreting the layout.  For example, I doubt the women were 207 y old, on average.  What was the mean age for men and women?  Also, “mean ± SD” serves as a column heading but is then listed further below as a value. 

Lines 141-143: Are these numbers cumulative deaths or independent numbers for each corresponding interval of time?  Please clarify.

Table 2: The table could be simplified by stating mean +SD in the legend and then deleted the column of repeated statements of mean +SD since all values are just that. 

Likewise for Table 3; once stated in the legend, mean +SD would not need to be repeated in the table.

Lines 162-164, Table 4: Place add the sample size (n).  Was it n=94 or, with mortalities, did it change for each time interval?

Lines 220-222, “Phase angle is associated with prognosis in patients admitted for hip fracture, independent of age and comorbidities, and can be positioned as a prognostic tool for mortality at 1, 3, 6, and 12 months.” As described by the authors, the ROC analysis was only done for the 12-mo mortality rate (lines 144-148).  I realize multiple regression involved mortality at each of the four time points but this leads to me question about the survival analysis.  How was the change in the sample (prior mortalities) accounted for?  How large are the samples sizes in Table 4, and if small, how much confidence can we have in the conclusions?

Author Response

Thank you for your feedback on our article "Phase angle as a predictor of mortality in older patients with hip fracture". I will be responding to your comments in this letter, giving reasons for our decisions and explaining the changes made in response to your suggestions. When referring to a line in the article, I will use the numbers corresponding to the new and revised version of the manuscript.

Major Concerns

Comments 1: The authors claim that no prior study has investigated the relationship between mortality and phase angle in those with hip fractures (line 72 and lines 166-167).  While that might be true for phase angle, 145 references came up in PubMed using a search on the words, mortality, hip fracture, elderly, and muscle.  For some of these studies, the measurement of muscle was more direct or specific than phase angle.  The authors have acknowledged the benefit of phase angle (i.e., unobtrusive and doesn’t require body mass assessment) but I think they could improve their acknowledgment of what has been done previously.

Response 1: Thank you for pointing this out. We have already included in the discussion of our article some references to articles that have examined and demonstrated the association between muscle mass and mortality in older people with hip fracture (lines 200-202). We are aware that this may have needed to be mentioned in the introduction. Therefore, in the revised version of the article, an introductory line has been added to the introduction section (lines 80-81).

Comments 2: Because of the longitudinal aspect, i.e., following mortality periodically over 12 mo, it isn’t clear whether the authors adjusted mortality rates for prior deaths.  If not, a survival analysis might be warranted.

Response 2: Thank you for your feedback. We want to clarify that in our analysis, we have indeed accounted for prior deaths at each follow-up point. By considering the cumulative mortality up to each time point, we ensure that the reported mortality rates reflect the impact of previous deaths accurately.

Specifics Comments/Questions 

Comments 3: Lines 23-24: Here or in a more appropriate place in the abstract, add that receiver operating curves were used to determine cutoffs for mortality.

Response 3: The statement has been added to the revised version of the manuscript (lines 24-25). In order to meet the 200-word limit for the abstract, minor changes have been performed.

Comments 4: Line 33, Key Words: Add “muscle mass” as a key phrase.

Response 4: The key phrase has been added to the revised version of the manuscript (lines 32-33).

Comments 5: Lines 93-94: Were the concentrations of these proteins measured in serum or plasma?  Please specify the methods.  Perhaps a reference or two or the source of the assay kits could be used here.

Response 5: They were measured in serum by using the following assay kits: Atellica® CH Albumin (Alb) and Atellica® CH Albumin BCP (AlbP) for albumin, Atellica® CH Prealbumin (PreAlb) for prealbumin and Atellica® CH C-Reactive Protein_2 (CRP_2) for CRP. This statement has been added to the revised version of the manuscript (lines 104-108).

Comments 6: Lines 101-107:  Please indicate/add how the researchers ensured patients arrived well hydrated and how long they remained in the supine position.  These factors impact accuracy or reliability of the BIA measurements.  With the assessment done 24 to 48 h post-surgery, a time when subjects may be in negative nitrogen balance, when any requirements for BIA use violated?

Response 6: To ensure correct hydration of the patients, our team thoroughly reviewed the post-surgical hydration protocols to ensure they were followed correctly. Possible interferences of solutions and other intravenous products were avoided at the time of the assessment. Patients adopted a supine position for at least 5 minutes. These clarifications have been added to the revised version of the manuscript (lines 116-120).

Comments 7: Lines 105-108: How was phase angle converted to fat-free mass?  In other words, what equation or constant was used to go from phase angle to fat-free mass?  Was the equation valid for this age group of males and females?  Again, a reference to the validation might suffice here.  What is the sample size for FFM and muscle mass?  Line 97 states “When possible, height was determined by using a stadiometer.”  How many were unable to participate in the stature measurement?

Response 7: Thank you for your insightful questions. To clarify, the phase angle was not directly converted to fat-free mass (FFM); instead, FFM was calculated using the proprietary equations provided by the Bioelectrical Impedance Analysis (BIA) device, which are based on resistance and reactance measurements, including the phase angle. We have added details about the obtention of the parameters in the Introduction and the Materials and Methods sections to provide further clarity (lines 68-71,74-78 and 121-125). The sample size for FFM and muscle mass measurements, as it is shown in table 2, includes all participants in the study, as we successfully obtained these measurements for everyone. In instances where it was not possible to measure height with a stadiometer, we used self-reported height. This approach ensured comprehensive data collection for all participants. Unfortunately, we are not able to provide with the number of participants whose height was not possible to measure and self-reported height had to be used.

Comments 8: Line 133, Table 1: The numbers seem to be misaligned for the column headings or perhaps I’m misinterpreting the layout.  For example, I doubt the women were 207 y old, on average.  What was the mean age for men and women?  Also, “mean ± SD” serves as a column heading but is then listed further below as a value. 

Response 8: The middle column was meant to explain how the variable on the left was displayed. Since the text in the row corresponding to the variable “age” was bolded, it may have misled to think that it corresponded to a heading row. To avoid further misunderstanding, the table has been edited, bolding the heading row and the name of the variables in the first column, and deleting the middle column, explaining in the footnote how each variable is displayed.

Comments 9: Lines 141-143: Are these numbers cumulative deaths or independent numbers for each corresponding interval of time?  Please clarify.

Response 9: They refer to cumulative deaths.  I hope that the edited version of this paragraph helps to understand it better (lines 160-163).

Comments 10: Table 2: The table could be simplified by stating mean +SD in the legend and then deleted the column of repeated statements of mean +SD since all values are just that.

Response 10: We agree with this comment. Moreover, the table has been edited to compare its variables between sexes, following advice from the other reviewer.

Comments 11: Likewise for Table 3; once stated in the legend, mean +SD would not need to be repeated in the table.

Response 11: That column has been removed from the revised version of the manuscript as well.

Comments 12: Lines 162-164, Table 4: Place add the sample size (n).  Was it n=94 or, with mortalities, did it change for each time interval?

Response 12: Thank you for your insightful questions regarding Table 4. To clarify, the logistic regression analysis was conducted using the entire study cohort of 262 patients. The sample size remains n=262 throughout the analysis. Although we present odds ratios and p-values at 1, 3, 6, and 12 months, these values reflect the association between low phase angle and mortality at each specific time interval independently. This approach ensures that the initial sample is consistently used to assess the odds of mortality at different follow-up points. We have added a note to Table 4 to indicate that the sample size for the logistic regression analysis remains n=262 throughout the study period.

Comments 13: Lines 220-222, “Phase angle is associated with prognosis in patients admitted for hip fracture, independent of age and comorbidities, and can be positioned as a prognostic tool for mortality at 1, 3, 6, and 12 months.” As described by the authors, the ROC analysis was only done for the 12-mo mortality rate (lines 144-148).  I realize multiple regression involved mortality at each of the four time points but this leads to me question about the survival analysis.  How was the change in the sample (prior mortalities) accounted for?  How large are the samples sizes in Table 4, and if small, how much confidence can we have in the conclusions?

Response 13: Thank you for your insightful questions. To clarify, we utilized a ROC curve analysis to determine the optimal phase angle cutoff for predicting 12-month mortality. This cutoff was then used to classify patients into two groups: normal and low phase angle. This dichotomous variable was subsequently used in our logistic regression analyses at each follow-up point (1, 3, 6, and 12 months). Each logistic regression analysis was conducted independently, thus ensuring that prior mortalities did not affect the sample size or results at subsequent intervals.

Regarding your concern about sample sizes, each logistic regression included the full initial cohort of 262 patients, irrespective of mortalities occurring at earlier time points. This approach ensures that our findings are robust and that the sample sizes for each time point are consistent. Therefore, we have confidence in the conclusions drawn from these analyses

We hope that the answers given and the changes made are appropriate and accurate.

Round 2

Reviewer 1 Report

Comments and Suggestions for Authors

no

Comments on the Quality of English Language

no

Reviewer 2 Report

Comments and Suggestions for Authors

The authors have addressed all of my concerns adequately and improved the understanding of the study.  I have no further questions or concerns.